# Catalyst-Free Cardanol-Based Epoxy Vitrimers for Self-Healing, Shape Memory, and Recyclable Materials

**DOI:** 10.3390/polym16030307

**Published:** 2024-01-23

**Authors:** Yu Zhu, Wenbin Li, Zhouyu He, Kun Zhang, Xiaoan Nie, Renli Fu, Jie Chen

**Affiliations:** 1Key Laboratory of Biomass Energy and Material, Jiangsu Province, Institute of Chemical Industry of Forest Products, Chinese Academy of Forestry, Nanjing 210042, China; 2College of Materials Science and Technology, Nanjing University of Aeronautics and Astronautics, Nanjing 210016, China

**Keywords:** bio-based vitrimers, cardanol, dynamic transesterification, catalyst-free, recyclability

## Abstract

Bio-based vitrimers present a promising solution to the issues associated with non-renewable and non-recyclable attributes of traditional thermosetting resins, showcasing extensive potential for diverse applications. However, their broader adoption has been hindered by the requirement for catalyst inclusion during the synthesis process. In this study, a cardanol-based curing agent with poly-hydroxy and tertiary amine structures was prepared by a clean synthetic method under the theory of click chemistry. The reaction of a cardanol-based curing agent with diglycidyl ether of bisphenol A formed catalyst-free, self-healing, and recyclable bio-based vitrimers. The poly-hydroxy and tertiary amine structures in the vitrimers promoted the curing of epoxy-carboxylic acid in the cross-linked network and served as internal catalysts of dynamic transesterification. In the absence of catalysts, the vitrimers network can achieve topological network rearrangement through dynamic transesterification, exhibiting excellent reprocessing performance. Moreover, the vitrimers exhibited faster stress relaxation (1500 s at 180 °C), lower activation energy (92.29 kJ·mol^−1^) and the tensile strength of the recycled material reached almost 100% of the original sample. This work offers a new method for preparing cardanol-based epoxy vitrimers that be used to make coatings, hydrogels, biomaterials, adhesives, and commodity plastics in the future.

## 1. Introduction

Vitrimers are a class of resins that change their cross-linked network structure through dynamic covalent bonds (DCBs). Unlike epoxy resins, which are widely used in life, the covalent adaptive networks formed by the DCBs endow vitrimers with certain recoverability [1,2]. Based on the thermodynamic equilibrium reaction, DCBs can undergo bond breaking and recombination under certain conditions (e.g., temperature, light, pH) [3], so that vitrimers have reprocessability similar to thermoplastic resins [4,5,6]. At the service temperature, they have the same stable crosslinking network as thermosets. Therefore, such epoxy vitrimers, which combine the advantages of thermosetting and thermoplastic resins, have attracted extensive attention from researchers [7,8].

On this basis, vegetable oil-based epoxy vitrimers in line with sustainable development strategies are being developed. Environmentally friendly and quickly degradable epoxy vitrimers can be obtained by introducing DCBs into the cross-linked network through reactions such as Diels–Alder adjunction, metathesis of benzyl cyclic acetal, disulfide, imine, hydroxyl ester bonds, and silyl ethers [9,10,11,12,13,14]. Vitrimer based on a dynamic transesterification system is widely used because of its simple preparation process and good performance. Dynamic ester linkages can be introduced into the cross-linked network by curing with carboxylic acids and epoxy groups. Carboxylic acids react with epoxy groups to form esters and hydroxyl groups, and transesterification can occur between the two groups, making ester bonds reversible [15,16,17]. Wu et al. described the curing of epoxidized soybean oil with glycyrrhetinic acid to prepare fully bio-based vitrimers with high thermal stability, shape-changing and repairing properties, which can be recycled and chemically degraded in ethylene glycol [18]. Zhang et al. reported that ozone-treated kraft lignin and sebacic acid epoxy were cured under the action of a zinc catalyst to form epoxy vitrimers with excellent shape memory and repairing properties, which can be used as recoverable adhesives [19]. Qi et al. reported that fatty-acid-based epoxy vitrimers, with rapid stress relaxation behavior at 180 °C, were catalytically formed from fatty acids and diglycidyl ether of bisphenol A. The vitrimers are 100% recyclable with almost no loss of material properties after recycling [20].

However, catalysts are required in the above transesterification systems and include zinc II [6], tin II [21,22] and tertiary amines (e.g., TBD [23], DBU [24]). Among them, many catalysts are moderately toxic, and unusable in daily life, and may even corrode the substrate. More importantly, there are also problems such as catalyst leaching and hydrolysis in vitrimers [25]. In the existing research, it has been confirmed that when the cross-linked network contains enough -OHs and tertiary amine structures which can form hydrogen bonds or ionic bonds with carboxylic acids or carboxylic acid derivatives, thereby reducing the activation energy of the dynamic transesterifications (DTERs), promoting the reaction [26,27]. Therefore, it is of great significance to develop some catalyst-free epoxy vitrimers.

Cardanol is a renewable resource extracted from natural cashew nutshell oil through advanced technology [28]. Owing to its easy modification and high flexibility, cardanol can partially replace phenol for the manufacture of epoxy curing agents [29,30,31]. Furthermore, because of its aromatic and aliphatic structure, cardanol can enhance the chemical and mechanical resistance, corrosion resistance and flexibility of epoxy coatings [32]. In addition, the meta-position of its phenol is replaced by a long straight chain containing 0 to 3 unsaturated bonds, which can be chemically modified to synthesize the epoxy monomer or epoxy curing agent. Generally, cardanol is mainly used as a modifier of epoxy resin, and a small amount of addition can achieve the effect of toughening [33,34]. However, cardanol-based epoxy vitrimers suffer from poor mechanical properties and are less studied [35,36].

In this work, an epoxy curing agent (CAPA, cardanol amine polyacid) with a tertiary amine structure and -OHs was prepared from renewable cardanol using a clean synthetic method based on the principles of click chemistry (Figure 1). Through the curing reaction between this curing agent and diglycidyl ether of bisphenol A (DGEBA), DTERs were introduced into the epoxy network. The long-chain structure of cardanol combines with the rigid benzene ring in DGEBA to form a rigid-flexible network. Even the tertiary amine structure in the curing agent allows DGEBA-CAPA to be cured without additional catalysts. At higher temperatures, DTERs between -OHs and ester bonds become active, which endows the materials with reparability and certain malleability [37]. Subsequently, the properties of the epoxy vitrimers are regulated by modulating the ratio of epoxy to carboxylic acid. Thus, the well-designed epoxy thermoset with DTERs was fabricated successfully. In addition, another epoxy curing agent (CDPA, cardanol polyacid) without tertiary amines or -OHs structures was synthesized from direct clicking, and used as a comparison to verify the effect of the designed structure on the performance of vitrimers. More importantly, there are few studies on cardanol-based epoxy vitrimers, and this research also provides a new idea for the synthesis of cardanol-based epoxy vitrimers.

## 2. Materials and Methods

### 2.1. Materials

Cardanol (CD, Shanghai Meidong Biomaterials Co., Ltd., Shanghai, China) was purified by vacuum distillation at 240 °C. Diglycidyl ether of bisphenol A (DGEBA, epoxide value: 0.51 mol·(100 g)^−1^) was obtained from Nantong Xingchen Synthetic Materials Co., Ltd. (Nantong, China). 3-mercaptopropionic acid (MPA), 2-Hydroxy-2-methyl-1-phenyl-1-propanone (photo initiator 1173) were purchased from Shanghai Macklin Biochemical Co., Ltd. (Shanghai, China). Epichlorohydrin (ECH), Diethanolamine (DEA), benzyl-triethylammonium chloride (TEBAC), 2,4,6-tris (diethylaminomethyl) phenol (DMP-30) were purchased from Shanghai Aladdin Bio-Chem Technology Co., Ltd. (Shanghai, China). 

### 2.2. Synthesis of Cardanol Glycidyl Ether

First, CD (0.17 mol, 51.34 g), ECH (1.32 mol, 122.13 g) and TEBAC (0.005 mol, 1.14 g) were added to a four-port flask and reacted at 117 °C for 2 h under N_2_. Next, when the temperature of the system was dropped to 60 °C, NaOH (0.17 mol, 6.6 g) was put into the flask and reacted for 3 h. Cardanol glycidyl ether was obtained through filtration and rotary evaporation, named CGE (epoxide value: 0.28 mol·(100 g)^−1^).

### 2.3. Synthesis of Cardanol Amine Polyol

CGE (0.1 mol, 35.6 g) and DEA (0.1 mol, 10.5 g) were added to a four-port flask equipped with a stirring paddle. The reaction needed to be performed at 70 °C for 4 h under N_2_. After the reaction, the upper light yellow organic phase was dissolved and extracted using ethyl acetate and subsequently washed with saturated saline and deionized water to neutral. Subsequently, anhydrous sodium sulfate was added to remove water. After night drying and filtration followed by rotary evaporation to remove the excess solvent, cardanol amine polyol was obtained, named CAP.

### 2.4. Synthesis of Cardanol Amine Polyacid

CAP (0.05 mol, 23.05 g), MPA (0.15 mol, 15.90 g) and photoinitiator 1173 (2 wt%, 0.8 g) were added into a quartz glass beaker. The mixture was stirred well and reacted for 12 h in the case of ultraviolet (UV)-light. The wavelength range of ultraviolet light is 300–350 nm. The goal products were obtained after removing excess MPA by rotary evaporation at 80 °C, named CAPA (acid value: 170 mg KOH·g^−1^). As shown in Appendix A, the synthesis method of CDPA refers to the synthesis method of CAPA, i.e., CD is used instead of CAP (acid value: 275 mg KOH·g^−1^).

### 2.5. Preparation of Epoxy Vitrimers

The abbreviations used in this article are shown in Table 1. Firstly, the acid value of the CAPA obtained above was measured and the ratio of DGEBA to CAPA was calculated. The ratio of DGEBA and CAPA calculated based on the acid value was 1:1, that is, the epoxy group: carboxyl = 1:1. Then, the ratio of DGEBA and CAPA was 1:1, 1:0.75, 1:0.5, and 1:0.25 for curing experiments, which were noted as DGEBA-1CAPA, DGEBA-0.75CAPA, DGEBA-0.5CAPA and DGEBA-0.25CAPA, respectively. For instance, the specific curing process for DGEBA-0.5CAPA is as follows: CAPA (16.8 g) and DGEBA (20 g) were added into the beaker and mixed well. Before pouring into the stainless-steel molds, all mixtures were degassed through a vacuum oven. The curing process was conducted at 120 °C for 2 h, 140 °C for 2 h, 160 °C for 4 h and 180 °C for 1 h. In addition, the same curing operation was performed with DGEBA-CDPA as a comparison experiment.

### 2.6. Characterization and Instruments

The chemical structures of the products were determined using Fourier transform infrared spectroscopy (FTIR) and ^1^H nuclear magnetic resonance (NMR). FTIR was performed using a Thermo Scientific Nicolet iS10 spectrometer (Waltham, MA, USA), covering a wavenumber range of 600–4000 cm^−1^. The ^1^H NMR spectra were recorded on a Bruker 400 MHz spectrometer (Billerica, MA, USA), employing deuterochloroform (CDCl_3_) as the solvent and tetramethylsilane (TMS) as the internal standard.

Differential scanning calorimetry (DSC) was performed using DSC 8000 (PerkinElmer, UK), and all mixtures were scanned from 30 to 180 °C at a heating rate of 10 °C·min^−1^. Dynamic Mechanical Analysis (DMA) was conducted using a DMA Q800 from TA Instruments (New Castle, DE, USA) in dual cantilever mode. The analysis was conducted at a constant oscillating frequency of 1 Hz. Samples measuring 60 mm × 10 mm × 4 mm were scanned over a temperature range of −50 to 180 °C, with a heating rate of 3 °C·min^−1^. Thermogravimetric Analysis (TGA) was performed using a Netzsch 409PC instrument (Selb, Germany). Under the nitrogen atmosphere, samples were heated from 30 to 800 °C at a rate of 20 °C·min^−1^. Stress relaxation was measured using a DMA Q800 from TA Instruments. During the test, 35 × 6 × 1 mm^3^ samples were given a constant strain of 1% and the relaxation modulus was recorded at 170–200 °C as a function of time. The mechanical properties of the epoxy vitrimer were confirmed using an Instron 4201 universal testing machine (Norwood, MA, USA) according to the ASTM D638-14 standard method [38]. Each sample was tested repeatedly at least five times. Surface scratch experiments for self-healing properties were carried out using an Optec BK-POL polarized light microscope (Cnoptec, Chongqing, China)and a DV500 digital camera (Cnoptec, Chongqing, China). Connect the microscope to the computer, select the picture and measure the width of the scratch using the software called S-Viewer (V1.20).

## 3. Results

### 3.1. Characterizations of the Synthesized Cardanol-Based Curing Agent

The ^1^H nuclear magnetic resonance (^1^H NMR) and Fourier transform infrared spectrometry (FTIR) spectra were used to determine the molecular structures of CD, CGE, CAP, and CAPA. The absorption peaks assigned to phenolic hydroxyl (3337 cm^−1^), and C=C (3008, 911, 873 cm^−1^) can be observed in the spectrum of CD (Figure 1) [39]. The absorption peak associated with phenolic hydroxyl (3337 cm^−1^) was lost in the reaction from CD to CGE, and the peak related to the epoxy group (910 cm^−1^) was seen instead. In the reaction from CGE to CAP, with the addition of DEA, the absorption peaks of hydroxyl groups reappeared. Moreover, the peaks of C=C at 3008 and 878 cm^−1^ disappeared and a broader peak of the carboxyl group peak appeared in the spectrum of CAPA, which indicates the successful preparation of CAPA.

Appendix A represent the ^1^H NMR spectra of CD, CGE, CAP, and CAPA, respectively. The spectrum in Appendix A shows an obvious signal at 5.2–5.4 ppm (15, 16, 18, 19), corresponding to the double bond structure of the middle segment in the side chain of cardanol. The signals at 5.7–5.8 ppm (21) and 4.9–5.0 ppm (22) correspond to the double bond structure at the end of the side chain in cardanol [39]. The signals at 3.2–3.3 ppm (9) and 3.8–3.9 ppm (8) in Appendix A correspond to the newly generated epoxy groups resulting from the epoxidation of phenolic hydroxyl group in cardanol. The appearance of new signals at 3.5–3.6 ppm (11, 12) and 3.6–3.7 ppm (10) in Appendix A proves that the ring-opening reaction between the epoxy group and diethanolamine in CGE produces a hydroxyl group. The reaction from CAP to CAPA is a click process, and almost no side reactions occur under the UV irradiation. As shown in Appendix A, the double bond signals at 5.2–5.4, 5.7–5.8, and 4.9–5.0 ppm almost completely disappeared, and the peak of carboxyl appeared in the infrared spectrum, which proves the success of all reactions.

DGEBA-CAPA was directly cured and synthesized from a ring-opening reaction. Appendix A shows the infrared spectrum of DGEBA-CAPA after curing. The characteristic peaks of the epoxy group (911 cm^−1^) and the carboxyl group (1707 cm^−1^) are absent from the spectrum. The characteristic peaks of ester group (1731 cm^−1^) and hydroxyl group (3377 cm^−1^) are present, which indicates the successful synthesis of the DGEBA-CAPA epoxy vitrimer matrix.

### 3.2. Mechanical Properties

Mechanical properties were tested to evaluate the practicability of the epoxy vitrimers (Figure 2). The epoxy vitrimers with different ratios of DGEBA to CAPA present different mechanical properties. The tensile strengths of DGEBA-1CAPA and DGEBA-0.75CAPA are both lower than 2 MPa, but the elongation at break is 95.8% and 160.9%, respectively, indicating these epoxy resins can be used as elastomeric materials. The main reason is that CAPA accounts for the majority of this composition and the long chain of cardanol increases its flexibility [32]. In addition, DGEBA-0.5CAPA exhibits good overall mechanical properties, with a tensile strength of 10.75 MPa and an elongation at break of 172.5%. The hydrogen bond formed by -OHs can act as sacrificial bonds to break before the breakage of samples, and consume a part of the energy, which has a toughening effect on the material [40]. This high elongation at break, relative to its lower tensile strength, makes it suitable for many application scenarios where high toughness and ductility are required. DGEBA-0.25CAPA showed high rigidity with tensile strength up to 57.80 MPa and elongation at a break of 4.2%, when DGEBA was excessive and CAPA provided fewer long-chain structures. At this ratio, it would be impossible to form a gel in theory even if the average functionality of CAPA is 3. In fact, the significant increase in tensile strength is mainly due to the reaction of polyhydroxyl groups with epoxy groups and the epoxy homopolymerization initiated by the tertiary amine structure, which introduces more rigid benzene ring structures into the cross-linked network. To confirm this reaction, two different exothermic peaks can be seen from the DSC thermograms of DGEBA-CAPA systems (Appendix A). It can be seen that the reaction temperature of epoxy group and carboxyl group is 146.7 °C, while the other reaction temperature is 161.6 °C. As a result, the mechanical properties required in the application can be obtained by adjusting the ratio of DGEBA to CAPA.

### 3.3. Thermal Stability

The thermal stability of the cured samples was tested via thermal gravimetric analysis (TGA) thermograms (Figure 3), and all relevant parameters were listed in Table 2. The thermal decomposition temperature of the cured epoxy vitrimers was distributed at 300–350 °C. Clearly, the vitrimers cured by cardanol derivatives exhibited excellent thermal stability, and their initial pyrolysis temperature (*Ti_%_*) was up to 349.7 °C. As the proportion of CAPA fraction increases, the thermal decomposition temperature of the epoxy vitrimers tends to fall. Owing to the increase in ester bonds, and the decomposition range of 300–450 °C is attributed to the breakage of the ester bond segment in the epoxy vitrimer structure, resulting in a decrease in thermal stability. The derivative thermogravimetry (DTG) curve shows two peaks in DGEBA-1CAPA: the peak at 330 °C represents the decomposition process of CAP without double bond structure, and the peak at 374.2 °C stands for the maximum degradation rate of normal epoxy vitrimers. Notably, the thermal stability of the vitrimers prepared here is higher than many reported vegetable oil-based epoxy thermosetting resins [11,41,42], which may be ascribed to the high-temperature resistance of the benzene ring structure of cardanol [32].

### 3.4. Dynamic Mechanical Properties

The microstructures of these materials were further investigated using dynamic mechanical analysis (DMA) to ascertain their dynamic mechanical properties. Figure 4 shows tan *δ* as a function of temperature for the resulting vitrimers. Among all components, DGEBA-0.25CAPA exhibited the highest *T_g_* (measured from the peak temperature of tan *δ*) ~57.8 °C. The *T_g_* value of the cross-linked polymer is affected by the stiffness of the skeleton and the cross-linking density. Excessive DGEBA provides a more rigid benzene ring structure for the cross-linked network, which hinders the movement of the molecular chain, so DGEBA-0.25CAPA exhibits the highest *T_g_*. The *T_g_* value of DGEBA-1CAPA is instead the lowest, mainly because more flexible chain structures of CAPA make the molecular chains move more easily.

To explain the above results, the cross-linking density (*ρ*) is calculated according to Formula (1):(1)$$\rho =\frac{E^{\prime }}{3RT}$$
where *R* is a universal constant (8.314 J·mol^−1^·K^−1^), *E*′ is the storage modulus at (*T_g_* + 30) °C and *T* corresponds to the absolute temperature of (*T_g_* + 30) °C. Consequently, the *ρ* at the different ratios of DGEBA to CAPA were easily determined to be 0.23 × 10^−3^, 0.21 × 10^−3^, 0.17 × 10^−3^ and 0.16 × 10^−3^ mol·cm^−3^, respectively (Table 3). Clearly, With the decrease in CAPA content, the epoxy-carboxyl crosslinking sites in the curing system reduced, so the crosslinking density showed a downward trend. The lower crosslink density is due to the dilution effect of its C_15_ alkyl side chain, which is consistent with the cardanol-based thermoset polymers studied by others [36]. The main reason for the difference in dynamic mechanical properties is that the rigid benzene ring structure of DGEBA and the flexible long chain of CAPA form a rigid-flexible network, while the different ratios of DGEBA to CAPA result in the difference in 3D network structure.

### 3.5. Stress Relaxation Behavior

The cross-linked network was rebuilt by thermally inducing DTERs in the network structure of epoxy vitrimers, which caused a stress relaxation behavior. According to the Maxwell model of relaxation response theory, the time at which the modulus relaxes to 1/e of the initial modulus, is considered the relaxation time (*τ**) of vitrimers. Figure 5a shows the stress relaxation performance at different ratios of DGEBA-CAPA at 180 °C. Clearly, the stress relaxation time is prolonged as the content of CAPA gradually decreases. The relaxation time of DGEBA-1CAPA is as short as 1530 s, while the relaxation time of DGEBA-0.25CAPA is longer at 180 °C.

The presence of -OHs in the network significantly affects DTERs, and the dynamic ester exchange between hydroxyl groups (-OHs) and ester bonds becomes more active at higher temperatures with an increased content of -OHs [37]. More importantly, the CAPA contains a tertiary amine structure, which also has a catalytic effect on DTERs in epoxy vitrimers [43]. In addition, DGEBA-0.5CDPA without the tertiary amine structure and -OHs showed no stress relaxation behavior at 180 °C for 10,000 s, while DGEBA-0.5CAPA showed a relaxation time of 3808 s, which also demonstrated the synergistic catalytic effect of the synthesised tertiary amine and -OHs on the DTERs (Figure 5b).

Subsequently, the effect of temperature on the stress relaxation of epoxy vitrimers was verified with DGEBA-1CAPA (Figure 5c). Obviously, the stress relaxation time of epoxy vitrimers is shorter with the increase in system temperature. The main reason is that the system temperature rises, the DTERs in the cross-linked networks of DGEBA-1CAPA and the topological network reorganization rate are accelerated, resulting in shorter stress relaxation times.

Especially, the transesterification ability can be expressed via activation energy by Arrhenius’ Equation (2) and Figure 5d as follows:(2)$$ln\tau =\frac{E_{a}}{RT}-lnA$$
where *τ* is the stress relaxation time; *E_a_* is the activation energy of the transesterification reaction; *R* is the gas constant; *A* is a pre-exponential factor. The activation energy of DGEBA-1CAPA is 92.29 kJ·mol^−1^, which is comparable to the data of vitrimers (69–150 kJ·mol^−1^) [2,8,44,45].

### 3.6. Self-Healing, Shape Changing and Recyclability

The cross-linked network induced by dynamic transesterification endows the vitrimers with self-healing properties. To evaluate this performance, the procedure was as follows: scratch a glass body with a smooth surface with a scalpel, and then heat it in an oven at 180 °C for different times to complete the thermally induced self-repair test, and record the size of the scratches at different repair times. Define the percentage of the repaired scratch to the original scratch size as the self-healing efficiency under this condition. Figure 6 shows the crack healing images heated at 180 °C for different time periods. DGEBA-1CAPA exhibited the best repair performance, and the cracks were almost completely recovered within 15 min. This is because the cross-linked networks of DGEBA-1CAPA contain abundant ester bonds, hydroxyl groups and tertiary amine structures, which can be reconstructed in the networks of vitrimers by DTERs at high temperatures. DGEBA-0.25CAPA exhibited the slowest self-healing rate, recovering to 42.9% of the crack width after 15 min, which is consistent with its slow stress relaxation rate. For comparison, due to the absence of hydroxyl groups and tertiary amine structures in CDPA, the rate of DTERs is relatively slow. After heating at 180 °C for 15 min, the crack width was only restored by 41.1%, which is much lower than that of DGEBA-1CAPA (Appendix A).

The remolding mechanism based on the transesterification exchange reaction is illustrated in Figure 7a, nucleophilic hydroxyl groups can react with the ester groups to form an associate intermediate at higher temperatures and then release the exchangeable hydroxyl groups and ester groups. As shown in Figure 7b, the sample was bent into a U-shape at 180 °C and kept for 2 h. Subsequently, the sample was removed and cooled slowly at room temperature. The cooled sample was then bent into an S-shape at 100 °C and cooled to room temperature, and a temporary S-shape was obtained. Finally, the sample regained the permanent U-shape at 100 °C. Apparently, the process demonstrates the good shape memory property in the vitrimers, which provides a good basis for the subsequent DGEBA-CAPA composite.

The detailed reprocessing process is shown in Figure 7c. To evaluate recyclability, the vitrimer samples were shredded and then hot-pressed at 180 °C and 15 MPa for 1 h. Recycled sheets were cut into rectangular samples (50 × 4 × 1 mm^3^) and used in tensile testing. The tensile strength of the recycled samples was used to evaluate recyclability. Recovery efficiency is defined as the ratio of the tensile strength of the recovered sample to that of the original sample.

As expected, DGEBA-CDPA formed fine fragments after hot pressing and could not be used, which is the same as the stress relaxation comparison diagram (Appendix A). For DGEBA-CAPA, the reprocessed vitrimers were uniform and transparent under light. As shown in Appendix A, the tensile strengths of reprocessed DGEBA-1CAPA and DGEBA-0.75CAPA are 1.47 MPa and 1.76 MPa, and the recovery efficiencies are 98.9% and 95.7%, respectively. The tensile strength after reprocessing DGEBA-0.5CAPA is 9.31 MPa, and the recovery efficiency is 86.6%. The rate of DTERs in the cross-linked system decreases as the CAPA content decreases, so the tensile strength of the reprocessed DGEBA-0.25 CAPA is 45.59 MPa and the recovery efficiency is only 75%. In addition, DGEBA-1CAPA was reprocessed several times, and the variation trends of stress and strain were shown in Figure 8. It can be clearly seen that the mechanical properties of vitrimers did not decrease significantly with the increase in the number of cycles, and the recovery efficiency still reached 98.1% even after the third cycle.

DGEBA-CAPA has high recyclability, which is due to the high content of reversible ester bonds and the abundance of -OHs and tertiary amine structures in the CAPA. The above result illustrates the synergistic catalysis of -OHs and tertiary amine structures on the DTERs effect. In addition, DGEBA-CDPA has only a small number of hydroxyl groups from the reaction of epoxy groups and carboxylic acids, which is not enough to promote the transesterification reaction, resulting in difficult recyclability.

## 4. Conclusions

Renewable cardanol was subjected to epoxidation reaction, addition reaction, and mercaptoene click reaction to obtain the CAPA with the structures of -OHs and tertiary amines. Furthermore, catalyst-free vitrimers were prepared from the reaction of DGEBA with CAPA. These materials exhibit high thermal stability, and the initial temperature of thermal decomposition is around 330 °C. The rigid-flexible networks of DGEBA and CAPA also provide the vitrimers with good mechanical properties. The tensile strength of DGEBA-0.5CAPA is 10.72 MPa and the elongation at break is 172.5%, indicating it is a good flexible body material. The tensile strength of DGEBA-0.25CAPA is 57.8 MPa, which is much higher compared with some reported cardanol-based epoxy vitrimers. The stress relaxation tests show that the numbers of -OHs and tertiary amines structures increase in the cross-linked network of vitrimers, and the catalytic effect on DTERs is more obvious at high temperatures. In addition, the DGEBA-CAPA vitrimers have good shape memory, self-healing and reprocessing properties. The scratches of DGEBA-1CAPA almost disappeared within 15 min at 180 °C. Subsequently, DGEBA-1CAPA was physically recovered, and the recovery rate was close to 100%. Hence, cardanol is a promising vegetable oil-based raw material for the preparation of epoxy vitrimers, and this study provides a new synthetic scheme for the synthesis of cardanol-based vitrimers.

## Data Availability

Data are contained within the article and Appendix A.

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
