# Peer review of "Catalyst-Free Cardanol-Based Epoxy Vitrimers for Self-Healing, Shape Memory, and Recyclable Materials"

_polymers, 2024, doi:10.3390/polym16030307_

Round 1
Reviewer 1 Report
Comments and Suggestions for Authors
This manuscript presents a new, catalyst-free, vegetable-oil-based epoxy vitrimer system, and demonstrates its key chemical and mechanical properties. I think it is a valuable contribution to the literature, but I will require a few modest changes before publication.
Regarding NMR results verifying the synthesis of CAPA, comment on formation of C-S bonds via secondary and tertiary proton integrals, not just loss of double bonds. I'm aware of the efficiency of thiol-radical click reaction, but it can't hurt to show explicitly.
The viscoelasticity information in table 2 is mislabeled with the thermal stability information from table 1.
Tensile strengths should be recorded above or below Tg to avoid rate dependence and have comparable data across specimens. You successfully achieved this with crosslink density and stress relaxation measurements, so it's not clear why tensile strength was omitted.
fig 6 methods are not even minimally described. They seem to be scratch healing micrographs but everything is labeled as crack healing. Clearly describe how scratches were introduced and how scratch widths were determined vs time such that the experiment can be reproduced. Scratch depth is also important for reproducibility. Likewise for fig S10. This omission is the only reason I selected major revision rather than minor.
Author Response
Response to Reviewer 1 Comments
Point 1: Regarding NMR results verifying the synthesis of CAPA, comment on formation of C-S bonds via secondary and tertiary proton integrals, not just loss of double bonds. I'm aware of the efficiency of thiol-radical click reaction, but it can't hurt to show explicitly.
Response 1: Thank you for your valuable feedback! From Figure S4 in the article, it can be seen that the double bond peak has completely disappeared and mercaptopropionic acid is excessive, so it can be proved that the reaction is complete. As can be seen from Figure 1 below, the content of cardanol without double bond structure is 3%, so the efficiency of this reaction is 97%.
Point 2: The viscoelasticity information in table 2 is mislabeled with the thermal stability information from table 1.
Response 2: Thank you for your valuable feedback! Modifications have been completed, please see Table 3 in the revised manuscript.
Point 3: Tensile strengths should be recorded above or below Tg to avoid rate dependence and have comparable data across specimens. You successfully achieved this with crosslink density and stress relaxation measurements, so it's not clear why tensile strength was omitted.
Response 3: Thanks for your comment! This experiment was conducted to measure the mechanical properties of vitrimer in accordance with the GB/T9431-2000 standard method. According to this standard, tensile property testing needs to be performed at 25°C.
Point 4: fig 6 methods are not even minimally described. They seem to be scratch healing micrographs but everything is labeled as crack healing. Clearly describe how scratches were introduced and how scratch widths were determined vs time such that the experiment can be reproduced. Scratch depth is also important for reproducibility. Likewise for fig S10.
Response 4: Thanks for your comment! The thermally induced self-repair performance of vitrimer was observed on an Optec BK-POL series polarizing microscope and digital photographs were taken with a DV500 camera. The procedure was as follows: scratch a glass body with a smooth surface with a scalpel, and then heat it in an oven at 180°C for different times to complete the thermally induced self-repair test, and record the size of the scratches at different repair times . Define the percentage of the repaired scratch to the original scratch size as the self-healing efficiency under this condition. Its interpretation has been added in the revised manuscript, please see 3.6 subsection in the revised manuscript.

Reviewer 2 Report
Comments and Suggestions for Authors
The work is of great quality on the subject of on cardanol-based epoxy vitrimers, and this research also provides a new idea for syn-91 thesis of cardanol-based epoxy vitrimers. however, to achieve significant improvements for its publication, I suggest the following modifications.
- In line 38, it is not advisable to start study in the middle of the introduction. I advise to remove this sentence.
- In “Materials and Methods” you need a table summarizing all samples manufactured duly followed with their respective abbreviations.
- In topic 2.6, it would be interesting to divide the tests separately.
Author Response
Response to Reviewer 2 Comments
Point 1: In line 38, it is not advisable to start study in the middle of the introduction. I advise to remove this sentence.
Response 1: Thanks for your comment! This sentence has been removed, please see Introduction in the revised manuscript.
Point 2: In “Materials and Methods” you need a table summarizing all samples manufactured duly followed with their respective abbreviations.
Response 2: Thank you for your valuable feedback. The annotation table has been added, please see Table 1 in the revised manuscript.
Point 3: In topic 2.6, it would be interesting to divide the tests separately.
Response 3: Thank you for your valuable feedback. Self-healing, shape changing and recyclability are all features of vitrimer, so they can be put together in the 3.6 subsection.

Reviewer 3 Report
Comments and Suggestions for Authors
1. Synthesis methodology is not clear
2. Purity degree?
3. NMR results?
4. English grammar is not good, it will revise
5. Catlyssi mechanism?
Comments on the Quality of English LanguageEnglish grammar is not good, it will revise
Author Response
Response to Reviewer 3 Comments
Point 1: Synthesis methodology is not clear
Response 1: Thanks for your comment! The specific synthesis scheme of CAPA has been clearly described, please see the ‘Materials and Methods’ part. The synthesis scheme of CDPA can be seen in 1.1 subsection in the supporting information.
Point 2: Purity degree?
Response 2: Thanks for your comment! From figure S4 in the supporting information, it can be seen that the double bond peak has completely disappeared and mercaptopropionic acid is excessive, so it can be proved that the reaction is complete. As can be seen from Figure 1 below, the content of cardanol without double bond structure is 3%, so the purity degree of CAPA is 97%.
Point 3: NMR results?
Response 3: Thanks for your comment! The NMR of all samples manufactured can be seen in the supporting information.
Point 4: English grammar is not good, it will revise
Response 4: Thank you for your valuable feedback. The English grammar has been revised, please see in the revised manuscript.
Point 5: Catlyssi mechanism?
Response 5: Thanks for your kind reminding! CAPA contains and hydroxyl structures. There are no active hydrogen atoms in tertiary amine, but there is still a lone pair of electrons on the nitrogen atom, which can carry out nucleophilic attack on the epoxy group, thus reducing the activation energy of the dynamic transesterification reaction and promoting the reaction. In addition, introducing more hydroxyl groups into the cross-linked system can indirectly shorten the distance between the ester bond and the hydroxyl group, thereby achieving the purpose of rapid transesterification.

Reviewer 4 Report
Comments and Suggestions for Authors
1. The authors must highlight the identified literature gap at the end of the introduction section.
2. Provide some SEM images on the fabricated vitrimers and tensile fractured surface SEM if possible
3. Compare the stress-strain values from tensile test before and after recycling.
4. Also include some SEM/optical images of recycled specimens
5. Quantify results in abstract.
Author Response
Response to Reviewer 4 Comments
Point 1: The authors must highlight the identified literature gap at the end of the introduction section.
Response 1: Thanks for your comment! In this work, an epoxy curing agent (CAPA, cardanol amine polyacid) with a tertiary amine structure and -OHs was prepared from renewable cardanol using a clean synthetic method based on the principles of click chemistry. In addition, there are few studies on cardanol-based epoxy vitrimers, and this research also pro-vides a new idea for synthesis of cardanol-based epoxy vitrimers. The introduction has been revised, please see in the revised manuscript.
Point 2: Provide some SEM images on the fabricated vitrimers and tensile fractured surface SEM if possible.
Response 2: Thanks for your kind suggestion. For materials like vitrimer, we pay more attention to its properties such as Self-healing, shape changing and recyclability. It is of little significance to take some SEM images on the tensile fractured surface SEM.
Point 3: Compare the stress-strain values from tensile test before and after recycling.
Response 3: Thank you for your valuable feedback. All temperature units are changed into degrees celsius, please see the first paragraph in the subsection 2.4 Characterization’ of the revised manuscript.
Point 4: Also include some SEM/optical images of recycled specimens
Response 4: Thank you for your valuable feedback. The answer is the same as point 4.
Point 5: Quantify results in abstract.
Response 5: Thanks for your kind suggestion. The abstract has been revised, please see the revised manuscript.

Reviewer 5 Report
Comments and Suggestions for Authors
An interesting manuscript, however, the author needs to revise it using the following comments:
1. Please make a separate Table mentioning the monomer content (in mol) during the preparation of your material.
2. 180 ℃ is too high, why did you choose this temperature for self-healing?
3. Please check the FT-IR and NMR of the materials after self-healing, and compare them before and after self-healing.
Comments on the Quality of English Language
Moderate revision is recommended.
Author Response
Response to Reviewer 5 Comments
Point 1: Please make a separate Table mentioning the monomer content (in mol) during the preparation of your material.
Response 1: Thanks for your comment. The content of CAPA and CDPA are calculated from its actual acid value, and the specific content of epoxy monomer and curing agent have been presented in a table. Please see the Table S1 in the supporting information.
Point 2: 180 ℃ is too high, why did you choose this temperature for self-healing?
Response 2: Thank you for your valuable feedback. The transesterification rate can be described by the stress relaxation time. As shown in Figure 5(a), the relaxation time of DGEBA-0.5CAPA at 180°C is 3780 s, while the modulus of DGEBA-0.25CAPA cannot relax to 1/e of the initial modulus even at 10000 s. Therefore, self-healing at 180°C is reasonable.
Point 3: Please check the FT-IR and NMR of the materials after self-healing, and compare them before and after self-healing.
Response 3: Thank you for your valuable feedback. The dynamic transesterification reaction occurs inside the cross-linked network, only a very small part of the bonds are broken and reorganized, and the FT-IR and NMR of vitrimer will not change significantly.

Round 2
Reviewer 1 Report
Comments and Suggestions for Authors
Response 1: Thank you for your valuable feedback! From Figure S4 in the article, it can be seen that the double bond peak has completely disappeared and mercaptopropionic acid is excessive, so it can be proved that the reaction is complete. As can be seen from Figure 1 below, the content of cardanol without double bond structure is 3%, so the efficiency of this reaction is 97%.
Follow-up 1: Please integrate the peaks between 3.0-2.5 and show in the main text that their sum is consistent with the proposed CAPA structure.
Response 3: Thanks for your comment! This experiment was conducted to measure the mechanical properties of vitrimer in accordance with the GB/T9431-2000 standard method. According to this standard, tensile property testing needs to be performed at 25°C.
Follow-up 3: It is valuable contribution to the overall literature data to measure to test standard and report that but, scientifically, the result is uninteresting because any differences between the materials' fundamental toughnesses at high and low temperatures are confounded by the interaction between the materials' Tgs and the strain rate of the tests. Consider augmenting the standard test with the pure tests I suggested.
Response 4: Thanks for your comment! The thermally induced self-repair performance of vitrimer was observed on an Optec BK-POL series polarizing microscope and digital photographs were taken with a DV500 camera. The procedure was as follows: scratch a glass body with a smooth surface with a scalpel, and then heat it in an oven at 180°C for different times to complete the thermally induced self-repair test, and record the size of the scratches at different repair times . Define the percentage of the repaired scratch to the original scratch size as the self-healing efficiency under this condition. Its interpretation has been added in the revised manuscript, please see 3.6 subsection in the revised manuscript.
Follow-up 4: This is mostly satisfactory, but please indicate in the main text specifically how the size of the scratches was measured. For instance, "The micrographs were calibrated against a reference X, giving pixel size Y. Images were processed with algorithm Z. The width of the scratch was manually measured at N points along the scratch using P% luminosity as the criterion for scratch/nonscratch pixels." or if a specific image analysis software was used, give the name, version, and relevant parameters.
Author Response
Point 1: Please integrate the peaks between 3.0-2.5 and show in the main text that their sum is consistent with the proposed CAPA structure.
Response 1: Thank you for your valuable feedback! From Figure S4 in the article, since the proton of the carboxyl group is affected by the electron-withdrawing effect of the two oxygens, the shielding is greatly reduced, so the chemical shift is 10-12 ppm. In addition, 2.5-3.0 ppm represents the signal of part of the hydroxyl group.
Point 2: It is valuable contribution to the overall literature data to measure to test standard and report that but, scientifically, the result is uninteresting because any differences between the materials' fundamental toughnesses at high and low temperatures are confounded by the interaction between the materials' Tgs and the strain rate of the tests. Consider augmenting the standard test with the pure tests I suggested.
Response 2: Thank you for your valuable feedback! Your question is very good and reasonable, and we have been actively looking for answers. However, we are sorry that the experimental facilities of our current research group cannot meet this experimental requirement. We are ready to continue exploring this in subsequent papers in the future.
Point 3: This is mostly satisfactory, but please indicate in the main text specifically how the size of the scratches was measured. For instance, "The micrographs were calibrated against a reference X, giving pixel size Y. Images were processed with algorithm Z. The width of the scratch was manually measured at N points along the scratch using P% luminosity as the criterion for scratch/nonscratch pixels." or if a specific image analysis software was used, give the name, version, and relevant parameters.
Response 3: Thanks for your comment! Your question is very good, and it is true that we did not express it clearly. We connect the microscope to the computer and use a software called S-Viewer (V1.20) to select pictures and use it to measure the width of the scratch. The procedure for measuring scratch width is described, please see in the Characterization and Instruments section.

Reviewer 3 Report
Comments and Suggestions for Authors
It is ok, just, check again before submission
Comments on the Quality of English Languageit is ok
Author Response
Thanks for your comment! The manuscript has been rechecked.
Reviewer 5 Report
Comments and Suggestions for Authors
Unfortunately, the authors didn't consider the suggestion, please consider the suggestion number 2 & 3.
For self-healing (response 1), 180 ℃ is too high. The authors can try at other temperatures, optimize, and then find the proper temperature for self-healing. The authors gave as a reference to choose that temperature based on Figure 5, but other temperatures less than 180 ℃ may work.
For the second point, the authors cured at 180 ℃ for self-healing, there is a chance of some other reaction/degradation. Only FT-IR/NMR can confirm the self-healing occurred only without any other new bond appearing.
In both responses, the authors mention some reason; but the experiment/analysis can only confirm these claims.
Revision needed
Author Response
Response to Reviewer 5 Comments
Point 1: For self-healing (response 1), 180 ℃ is too high. The authors can try at other temperatures, optimize, and then find the proper temperature for self-healing. The authors gave as a reference to choose that temperature based on Figure 5, but other temperatures less than 180 ℃ may work.
Response 1: Thanks for your comment. This is a very good and reasonable question of yours. Based on Maxwell's equations, we determined the topological freezing transition temperature (Tv) of the DGEBA-1CAPA network structure to be 159°C, and Tv represents the initial temperature at which vitrimer undergoes dynamic transesterification. The calculation process is as follows:
In the formula::
ƞ:1012 Pa·s-1ï¼›
τ*:Stress relaxation time, sï¼›
G:Shear modulus, Pa.
In the formula::
G:Shear modulus, Pa;
E :Average modulus of DGEBA-1CAPA samples in the temperature range 140 to 200 ℃, MPa;
ν:0.5ï¼›
Among them, E' is 1.7 MPa. The detailed calculation process is as follows:
G = 1.7/(2×1.5) = 0.57×106 Paï¼›
τ* = Æž/G = 1012/(0.57×106) = 1.8×106 sï¼›
lnτ* = ln (1.8×106) = 14.4ï¼›
In figure 5, y = 11.10x -11.31,14.4 = 11.10x-11.31,x = 2.32;
1000/Tv = 2.32,
Tv = 431.8 K ≈ 159 ℃.
In addition, the activation energy of DGEBA-0.5CAPA is higher, so its Tv should be higher than 159 ℃, so it is appropriate to choose 180 ℃ as the self-healing temperature. In other literature, the self-healing temperature of vitrimer based on transesterification reaction is also 180 ℃[1-6].
- Zhang, S.; Liu, T.; Hao, C.; Mikkelsen, A.; Zhao, B.; Zhang, J. Hempseed Oil-Based Covalent Adaptable Epoxy-Amine Network and Its Potential Use for Room-Temperature Curable Coatings. ACS Sustainable Chemistry & Engineering 2020, 8, 14964-14974.
- Li, W.; Xiao, L.; Huang, J.; Wang, Y.; Nie, X.; Chen, J. Bio-based epoxy vitrimer for recyclable and carbon fiber reinforced materials: Synthesis and structure-property relationship. Composites Science and Technology 2022, 227.
- Xu, Y.-z.; Fu, P.; Dai, S.-l.; Zhang, H.-b.; Bi, L.-w.; Jiang, J.-x.; Chen, Y.-x. Catalyst-free self-healing fully bio-based vitrimers derived from tung oil: Strong mechanical properties, shape memory, and recyclability. Industrial Crops and Products 2021, 171.
- Wu, J.; Yu, X.; Zhang, H.; Guo, J.; Hu, J.; Li, M.-H. Fully Biobased Vitrimers from Glycyrrhizic Acid and Soybean Oil for Self-Healing, Shape Memory, Weldable, and Recyclable Materials. ACS Sustainable Chemistry & Engineering 2020, 8, 6479-6487.
- Hao, C.; Liu, T.; Zhang, S.; Liu, W.; Shan, Y.; Zhang, J. Triethanolamine-Mediated Covalent Adaptable Epoxy Network: Excellent Mechanical Properties, Fast Repairing, and Easy Recycling. Macromolecules 2020, 53, 3110-3118.
- Yan, X.; Liu, T.; Hao, C.; Shao, L.; Chang, Y.-C.; Cai, Z.; Shang, S.; Song, Z.; Zhang, J. Rosin derived catalyst-free vitrimer with hydrothermal recyclability and application in high performance fiber composite. Industrial Crops and Products 2023, 202.
Point 2: For the second point, the authors cured at 180 ℃ for self-healing, there is a chance of some other reaction/degradation. Only FT-IR/NMR can confirm the self-healing occurred only without any other new bond appearing.
Response 2: Thank you for your valuable feedback. As shown in Figure 1, there is no obvious change in the FT-IR spectra of DGEBA-1CAPA before and after self-healing, proving that no degradation reaction has occurred.

Round 3
Reviewer 5 Report
Comments and Suggestions for Authors
The revision has been done accordingly, and it is recommended for publication
Comments on the Quality of English LanguageMinor revision is recommended.